# Combinatorial Virtual Library Screening Study of Transforming Growth Factor-β2–Chondroitin Sulfate System

**DOI:** 10.3390/ijms22147542

**Published:** 2021-07-14

**Authors:** Nehru Viji Sankaranarayanan, Balaji Nagarajan, Umesh R. Desai

**Affiliations:** 1Institute for Structural Biology, Drug Discovery and Development, Virginia Commonwealth University, Richmond, VA 23219, USA; nvsankaranar@vcu.edu (N.V.S.); bnagarajan@vcu.edu (B.N.); 2Department of Medicinal Chemistry, Virginia Commonwealth University, Richmond, VA 23298, USA

**Keywords:** glycosaminoglycans, chondroitin sulfate, growth factors, TGF-β2, molecular dynamics, molecular modeling, protein–ligand interactions

## Abstract

Transforming growth factor-beta (TGF-β), a member of the TGF-β cytokine superfamily, is known to bind to sulfated glycosaminoglycans (GAGs), but the nature of this interaction remains unclear. In a recent study, we found that preterm human milk TGF-β2 is sequestered by chondroitin sulfate (CS) in its proteoglycan form. To understand the molecular basis of the TGF-β2–CS interaction, we utilized the computational combinatorial virtual library screening (CVLS) approach in tandem with molecular dynamics (MD) simulations. All possible CS oligosaccharides were generated in a combinatorial manner to give 24 di- (CS02), 192 tetra- (CS04), and 1536 hexa- (CS06) saccharides. This library of 1752 CS oligosaccharides was first screened against TGF-β2 using the dual filter CVLS algorithm in which the GOLDScore and root-mean-square-difference (RMSD) between the best bound poses were used as surrogate markers for in silico *affinity* and in silico *specificity*. CVLS predicted that both the chain length and level of sulfation are critical for the high affinity and high specificity recognition of TGF-β2. Interestingly, CVLS led to identification of two distinct sites of GAG binding on TGF-β2. CVLS also deduced the preferred composition of the high specificity hexasaccharides, which were further assessed in all-atom explicit solvent MD simulations. The MD results confirmed that both sites of binding form stable GAG–protein complexes. More specifically, the highly selective CS chains were found to engage the TGF-β2 monomer with high affinity. Overall, this work present key principles of recognition with regard to the TGF-β2–CS system. In the process, it led to the generation of the in silico library of all possible CS oligosaccharides, which can be used for advanced studies on other protein–CS systems. Finally, the study led to the identification of unique CS sequences that are predicted to selectively recognize TGF-β2 and may out-compete common natural CS biopolymers.

## 1. Introduction

Transforming growth factor-beta (TGF-β), a member of the TGF-β superfamily, is known to play important roles in multiple diseases including various cancers (e.g., colorectal [1], glioblastoma [2], and squamous cell carcinoma [3]), neurodegenerative disorders (e.g., amyotrophic lateral sclerosis [4]), and tissue fibrosis (e.g., skeletal muscle fibrosis [5]). TGF-β is secreted by a large number of cells, although bone is its largest source. It is relevant to so many diseases because of its fundamental influence on cell growth and differentiation. Of specific importance to cancer, TGF-β has been shown to play both tumor-progression and -suppression roles depending upon the stage of disease [6], which makes it a challenging anti-cancer target [7].

Three isoforms of TGF-β (named TGF-β1, 2, and 3) exist in humans, with a high homology between them (~60–80%). The isoforms are 112-amino-acids long and primarily exist as homo-dimers, although hetero-dimers are possible [8]. The three-dimensional structures of TGF-β isoforms present a very interesting fold, which can be imagined as a ‘palm and fingers’ fold [9,10,11,12,13,14,15]. Dimerization on the ‘palm’ face with inverted symmetry yields the two ‘fingers’ domains in an extended geometry. Though the isoforms display a very high 3D similarity, differences in loop conformations and flexibility have been noted and proposed to be of functional significance [14].

Most members of the TGF-β superfamily bind to glycosaminoglycans (GAGs) [16,17], although much remains to be understood with regard to the specific role of this interaction. Based on commonalities observed with other growth factor family members, it can be minimally expected that GAGs present in the extracellular matrix serve as a reservoir of TGF-β. GAGs are also known to contribute to the formation of morphogen gradients, wherein proteins of the TGF-β family play an important role. Likewise, GAGs may modulate monomer–dimer equilibrium, thereby modulating cell surface receptor signaling in a spatiotemporal manner.

A handful of biophysical studies have been performed on GAG binding to TGF-β proteins. Of the three isoforms, TGF-β1 and TGF-β2 have been the most studied and found to interact with heparin/heparan sulfate (Hp/HS), chondroitin sulfate (CS), and sulfated hyaluronan (sHA) [18,19,20,21,22]. Based on these studies, higher levels of sulfation aided by longer chain lengths are two features that engineer higher affinities for TGF-β [16,17], although the TGF-β1–HS system has been reported to not follow the oligosaccharide size dependence [19,20]. Interestingly, our prior work on the CS binding to TGF-β2 present in human milk revealed a novel biphasic interaction mode [21]. More specifically, capillary electrophoretic studies have supported a dual-interaction model, wherein CS binds to TGF-β2 in two sites with different affinity values.

Despite the importance of TGF-β in human biology and the high possibility of the growth factor’s modulation by GAGs present in the extracellular matrix, atomistic details on the nature of the TGF-β interaction with GAGs are sparse. Crystallography studies not reported, and detailed computational studies are also lacking. In this work, we performed computational studies on the TGF-β2 monomer binding to CS (di- to hexa-saccharide long) to elucidate the key principles of recognition. This work also presents the generation of the in silico library of all possible CS oligosaccharides for the first time. We also demonstrate that the combinatorial virtual library screening (CVLS) algorithm, developed earlier to parse selective and non-selective Hp/HS sequences [23,24,25,26,27,28], can be equally implemented for CS02 (di-), CS04 (tetra-), and CS06 (hexa-) sequences. Finally, our exhaustive computational results support the biophysical studies-based biphasic interaction model reported earlier [21] and bring forth possible avenues for antagonizing or out-competing the biopolymer CS–TGF-β2 system.

## 2. Results

### 2.1. Application of CVLS Strategy to CS Sequences

The dual-filter CVLS strategy, first implemented in 2006 for Hp/HS sequences binding to antithrombin [23] and now shown to work for several other unrelated proteins [26,29], is a powerful strategy to answer fundamental questions regarding protein recognition by GAGs. Can CVSL be developed for CS sequences too? A key hypothesis of CVLS is that protein recognition by GAG sequences can be simulated within a short timespan if saccharide ring puckers and inter-glycosidic torsional angles (Φ and Ψ, respectively) fluctuate within a reasonably small range. If so, both puckers and torsions can be simulated in their “average conformations,” which dramatically reduces computational time, thereby enabling the screening of thousands of GAG sequences. This was found to be the case for Hp/HS sequences [23,26]. To assess whether this is true for CS sequences, we analyzed the reported crystal [30,31] and various theoretical structures [32] for CS and found that both GlcA*p* and GalNAc*p* residues adopt the characteristic ^4^C_1_ chair conformation of *D*-hexopyranoses. Likewise, the available data for inter-glycosidic torsions also indicate fairly consistent Φ and Ψ values of −81 ± 12 and −129 ± 14, respectively, for GlcA*p*–GalNAc*p* sequences and −69 ± 14 and 131 ± 9, respectively, for GalNAc*p*–GlcA*p* sequences [33]. Thus, these ring puckers and torsions were adopted in the generation of the library of all CS sequences.

The libraries of CS tetrasaccharides and hexasaccharides were generated from all possible monosaccharides, which included combinations of both natural and rare possible modifications of sulfation. In addition, sequences could have unsaturated uronic acid residue (ΔUA*p*), glucuronic acid (GlcA*p*), or galactosamine (GalN*p*) at the non-reducing end (NRE), which were named the ΔUA_NRE_, GlcA_NRE_, and GalN_NRE_ libraries, respectively (Figure 1). Using an automated procedure, a total of 192 (=64 × 3) tetrasaccharides (CS04) and 1536 (=512 × 3) hexasaccharides (CS06) were generated in a combinatorial manner.

For the CS02 and CS04 libraries, which consist of <500 sequences, a triplicate docking protocol was directly implemented and analyzed for high GOLDScore (“high-affinity”) and low RMSD (“high consistency of binding = selectivity”) binders using the top six poses of three independent docking runs. For the CS06 library, we followed the two-step CVLS protocol (Figure 1), which involved screening the entire library for docking onto the BS1 and BS2 of TGF-β2, identifying the top 2% of sequences from each library, and then more exhaustively performing docking in three independent runs and analyzing the top two docked poses from each run for RMSD. When the top six poses displayed an RMSD of <2.5 Å, the sequence(s) were deemed selective for recognition by TGF-β2.

The potential sites of CS binding were identified by searching for the presence of the Cardin and Weintraub sequence motif (Appendix A) [34] and electropositive domains through surface electrostatic potential calculation using the APBS tool in PyMol. Upon analyzing both the sequence and structure of TGF-β2, the presence of a Cardin and Weintraub pattern was not identified. Based on the electrostatic potential map, two potential binding regions (Figure 2) were identified: binding site 1 (BS1)m with Lys25, Arg26, Lys31, His34, Lys37, Lys94, and Lys97, and binding site 2 (BS2), with His58, Arg60, and Lys110.

### 2.2. CS Disaccharides Bind to TGF-β2, Albeit with Weak Consistency of Interactions

The library of CS02 disaccharides, consisting of 24 unique structures, was docked onto both putative sites of binding (BS1 and BS2) using GOLD (Figure 2). Typically, GOLD starts with a population of 100 arbitrarily docked ligand orientations, evaluates them using a scoring function (the GA “fitness” function), and improves their average “fitness” by an iterative optimization procedure that is biased towards high GOLDScores. For disaccharides, the parameters used were a 12 Å radius, 100 GA runs, and 100,000 iterations each. As shown in Table 1 and Figure 3, three and one sequence(s) bound consistently, i.e., RMSD < 2.5 Å, in BS1 and BS2, respectively. These sequences displayed GOLDScores ranging from ~40 to 56, which were relatively low in comparison to many literature reports [23,24,26,28,29]. Overlaying the docked poses showed that although their intra-sequence RMSDs were low, the binding orientations were varied (Figure 3). Similar conclusions could be drawn on the basis of the interactions with TGF-β2 residues. These findings indicated that TGF-β2 can bind to CS disaccharides, but the interaction is neither high-affinity nor selective.

### 2.3. Distinct CS Tetrasaccharides Bind TGF-β2 with Moderate Consistency

We studied all 192 tetrasaccharide sequences using the exhaustive screening parameters within CVLS (14 Å radius). Figure 4 shows the docked poses for the best CS04 sequences binding in either BS1 or BS2, and Table 2 lists their structures and CVLS parameters. Of the 192 sequences, three and seven CS04 sequences consistently bound in BS1 and BS2, respectively. The overlay of the three BS1-binding sequences revealed good consistency in terms of orientation and interactions. For the BS2-binding sequences, the overlay showed that two different binding modes, which alluded to the possibility of a moderate level of consistency. Additionally, more CS4 sequences were found to bind in BS2 than CS02 sequences, which, in our experience, is an unusual phenomenon because smaller sequences typically tend to be more easily accommodated. The GOLDScore for these preferred CS04 sequences binding to BS1 or BS2 were in the range of 45–62, which was in the range of that observed for the preferred CS02 sequences. Thus, we predicted that both sites, BS1 and BS2, may recognize longer sequences better.

### 2.4. A Small Group of CS Hexasaccharides Bind in BS1 and BS2 with High Consistency

To assess whether longer CS chains bind to TGF-β2 better, we screened the CS06 library of 1536 sequences in two steps, as shown in Figure 1. First, the entire library was docked onto both sites, BS1 and BS2, to identify the highest scoring ~2% sequences, which were then subjected a refined triplicate docking protocol for the analysis of consistency of binding. Table 3 and Figure 5 show the results obtained at both sites. For BS1, only one CS06 sequence displayed a high consistency of binding (RMSD ≤ 2.5 Å). An analysis using LigPlot+ showed the formation of multiple H-bonds with Arg26, Lys25, Lys37, Glu35, and Tyr91 residues, which contributed to a much improved GOLDScore of 88—nearly two-fold higher than that of CS04 sequences. For BS2, eight sequences consistently bound TGF-β2m with a high GOLDScore (70–88) and low RMSD of ≤2.5 Å. These sequences interacted with the His58, Ala74, Ser80, Asn103, Lys110, and Ser112 of BS2. Thus, only a small group of CS hexasaccharides bind in BS1 and BS2 with a high consistency. An important corollary in this connection is that the numbers of sulfates present in sequences do not correspond to higher selectivity; rather, a wider distribution of sulfate groups in a sequence corresponds to a higher selectivity. For example, the most sulfated CS sequence in the GlcA2S–GalNAc4S6S–GlcA2S–GalNAc4S6S–GlcA2S–GalNAc4S6S library did not display any selectivity in binding to TGF-β2.

### 2.5. MD Simulations Indicate Excellent Stability of the TGF-β2–CS06 Complexes in Both Sites of Binding

To assess whether the CS06 sequences identified through the CVLS algorithm display consistent and stable interaction with TGF-β2, we performed MD studies in a box of water under NPT conditions, as described earlier in our works for chemokine–GAG complexes [33,35]. The most optimal CS06 sequences binding to either BS1 or BS2 were used for MD studies. Initial runs were performed to ascertain that an MD simulation time of 25 ns was sufficient, which was in line with our earlier works [35,36].

The MD trajectory for BS1 complexation showed that both TGF-β2 and the bound CS06 reached an equilibrium well within 25 ns (see Appendix A). The RMSD fluctuations were within 2.90 and 2.83 Å for the protein and bound ligand, respectively, from the starting structure. The end-to-end distance (EED) of CS06 was stable across the MD trajectory and averaged 27.7 ± 1.5 Å (Appendix A). Similarly, the analysis of the MD trajectory for the CVLS-identified specific sequence binding in BS2 showed that both the protein and ligand were stable (RMSD of 3.18 and 2.37 Å, respectively; Appendix A). The EED of the ligand exhibited a high consistency (27.6 ± 1.6 Å) after achieving equilibrium (Appendix A).

### 2.6. A Combination of Direct and Water-Mediated Interactions Stabilize CS06 Binding in BS1

To understand the origins of the stability of the TGF-β2–CS06 complexes, we analyzed the inter-molecular hydrogen bond (H-bond) formation between protein and ligand atoms throughout MD trajectory. The CPPTRAJ tool, with H-bond donor–acceptor distance and angle cut offs of 3.0 and 135°, respectively [37], was used to identify direct H-bond formation in each MD frame, from which H-bond occupancy across the entire MD run was calculated and plotted using the in-house scripts. Likewise, the number of bridging water molecules between CS06 and TGF-β2 were also calculated using in-house scripts.

Figure 6A,B shows the interacting residues and their corresponding occupancies, respectively, for BS1. As one would expect on the basis of first principles, basic residues including Lys25, Lys31, Lys37, and Lys94 played a dominant role in binding to CS06. Interestingly, although Arg26 is located within BS1, it displayed a rather poor interaction with CS06 due to high gyrational motion during the simulation (see Appendix A). Another interesting observation was that BS1 could be thought of as a composite of two sub-sites formed by a helix/loop region and a beta-sheet region (see Appendix A), which bind the two terminal ends of the CS06 sequence with minimal interactions between TGF-β2 and the middle two saccharide rings of CS06. Alternatively, the middle two sugar rings appeared to ‘bridge’ the two sub-sites (Figure 7A). Such ‘bridging’ phenomenon is widely known to be important in the formation of protein–GAG–protein ternary complexes [27,38,39,40]. However, the occurrence of such a ‘bridging’ GAG chain that holds two disparate domains within the same protein has not been documented to the best of our knowledge. In fact, GAGs, especially Hp and HS, have been suggested to bind to a contiguous series of electropositive residues, e.g., Cardin–Weintraub sequences [34] or variants thereof [41,42,43].

Because sulfated GAGs are engulfed in water molecules, it was also important to assess their role in stabilizing the TGF-β2–CS complex. Figure 6C shows the H-bond occupancy of bridging water molecules interacting with residues of BS1. Interestingly, the positively charged His34 residue was found to bridge to beta sheet B for almost 1/3rd of the simulation time, suggesting its relevance in GAG recognition.

In addition to basic residues, several non-ionic residues of BS1 including Gly93, Tyr91, and Thr95 exhibited significant interactions with the CS06 sequence. In fact, the Gly93 amide atoms showed strong direct H-bonds with CS06, while main and side chain atoms of Thr95 contributed binding forces. The observation of these non-ionic interactions for the CS sequence revealed similarities with protein recognition of Hp/HS sequences. However, it is not clear as yet whether these non-ionic binding forces determine the selectivity of recognition, as has been observed for certain Hp/HS-binding proteins [44].

### 2.7. Role of Direct and Water-Mediated Interactions for BS2

The analysis of BS2 was performed in the same manner described above for BS1. Unlike the BS1 complex, the most optimal CS06 sequence bound in BS2 showed some translational drift within the H3 helix, though the interacting residues remained the same throughout the simulation (not shown). Furthermore, the direct and water-mediated H-bond interactions arose more from a diverse group of residues, including Gln57, His58, Ser80, Asn103, and Lys110, thus suggesting that recognition does not heavily rely on basic residues. In fact, the non-ionic residues Gln57, Ser80, and Asn103 were found to contribute more than Lys110 and His58, two important basic residues (Figure 7). This implies that CS06 recognition in BS2 is fundamentally different from that in BS1. More specifically, the selectivity of CS06 recognition in BS2 is slightly better than in BS1.

### 2.8. Simultaneous Engagement of BS1 and BS2 Results in Higher Stability of TGF-β2–CS06 Complex

While CVLS studies lead to predictions on recognition at individual sites, MD affords a fine opportunity to assess the simultaneous occupation of both sites. Thus, the MD of the TGF-β2–CS complex carrying two CS06 sequences bound in their respective sites, BS1 and BS2, was performed in a manner similar to that for individual sequences. The RMSD fluctuations for the protein averaged 2.96 Å, while those for the ligand averaged 3.29 Å (BS1) and 1.70 Å (BS2) (Appendix A). This was a striking result because the CS06 sequence in BS2 displayed much reduced fluctuations in the presence of the BS1 sequence than in its absence. This was also evident from the EED measurements (Appendix A), which displayed average values of 27.7 ± 2.1 and 28.8 ± 1.2 Å for CS06 in BS1 and BS2, respectively.

To assess whether the above-mentioned characteristics translate into higher stability, we performed free energy calculations on each MD frame (see Materials and Methods sections) of the protein–ligand co-complexes when CS06 was individually bound in either BS1 or BS2, as well as simultaneously in both sites. Figure 8 shows the total in silico free energy obtained from the MM-GBSA of the three co-complexes. While the TGF-β2–CS06 complex for the occupation of the BS1 site displayed a predicted energy of −27.4 ± 7.3 kcal/mol, that for the BS2 site was calculated to be −15.4 ± 13.4 kcal/mol. This implied that the two sites not only differentially recognize the two CS06 sequences but also differentially stabilize the chains. More specifically, the interactions of the BS2 residues were found to be weaker than those of the BS1 residues. This correlated well with experiments performed on the interaction of natural CS biopolymers with TGF-β2, which showed two binding sites with measurable differences in the solution affinities [21].

The results on the simultaneous occupation of both sites were more interesting. The predicted free energy increased more than two-fold to −69.8 ± 13.7 kcal/mol (Figure 8). To our knowledge, this is the first computational report showing the dual site engagement of proteins by GAGs, especially CS06. To ascertain that the system had reached equilibrium within 25 ns, we increased the simulation time to 50 ns and found a total binding free energy of −69.1 ± 15.8 kcal/mol, which was essentially identical to that for the shorter MD run (Figure 8).

### 2.9. Origin of Higher Stability for Dually Occupied TGF-β2 by CS06 Chains

To elucidate the atomistic origin of higher stability, we calculated direct and water-mediated H-bond occupancies, as well as the predicted single residue level energy contributions (SRED) of residues in BS1 and BS2 when simultaneously occupied (Figure 9). Few differences stood out. Arg26 and Tyr91 of BS1 and BS2, respectively (Appendix A), displayed significantly enhanced direct H-bonding in dually occupied TGF-β2. In fact, the occupancy of these two residues were found to be the among the highest for all residues (Figure 9B). This suggests that the conformational reorganization of TGF-β2 side chains induced a better fit. An identical conclusion could be drawn from the water-mediated H-bonding results, which showed that more robust interactions were engineered for most residues of BS1 and BS2, especially for Ser75 (BS1) and Lys94 (BS2) (Figure 9C), which were found to be absent in singly occupied TGF-β2.

In terms of SRED, Arg26 of BS1 was found to be a dominant contributor to binding energy in the dually occupied TGF-β2–CS06 complex. This residue, located in the H2 helix, was not found to play much of a role in the singly occupied system (Figure 6D). However, a large number of residues distributed over the entire BS1 and BS2 sites contributed to the stability of the dually occupied TGF-β2, which appeared to be the primary reason for the significant stabilization of the dually occupied TGF-β2–CS06 complex (Figure 9).

## 3. Materials and Methods

### 3.1. Software

SybylX 2.0 (Tripos Associates, St. Louis, MO, USA) was used for the molecular visualization, minimization, and preparation of protein structures from the Protein Data Bank (www.rcsb.org). GOLD, v5.8 [45], was used for molecular docking experiments. GAG sequences were combinatorially built in an automated manner using in-house SPL (Sybyl Programming Language) scripts and optimized in Sybyl.

### 3.2. Generation of Library of Chondroitin Sulfate Sequences

The first step in the CVLS approach is the generation of libraries that have a ΔUA*p, GlcAp,* or GalN*p* residue at the NRE so as to give the ΔUA_NRE_, *GlcA*_NRE_, and GalN_NRE_ libraries, respectively (Figure 10). ΔUA*p*/GlcA*p* (in ^4^C_1_ ring pucker) or GalN*p* (in ^4^C_1_) residues were combinatorially concatenated in the desired manner to generate each library of the desired chain length (i.e., di-, tetra-, or hexa-saccharide). The co-ordinates for the two libraries were generated in an automated fashion with a series of SPL scripts and a set of 24 disaccharide building blocks belonging to the ΔUA*p*–GalN*p* (Figure 10A), GlcA*p*–GalN*p* (Figure 10B), or GalN*p*–GlcA*p* (Figure 10C) series [46]. Herein, the different monosaccharide units were substituted with *O*-sulfate groups at appropriate places [46] to generate unique sequences. To name each unique CS sequence, the symbolic representation employed in the GLYCAM [47] designation was used. Briefly, the letter “Z” was used for GlcA*p*, “UA” for ΔUA*p,* and “V” for GalN*p*. Similarly, ring conformations were encoded as “b” for ^4^C_1_ conformations [48,49]. Substituents on rings were represented as follows: “C” (for *N*-acetyl), “2” (for *O*-sulfate), “4” (for 4-*O*-sulfate), and “6” (for 6-*O*-sulfate). The anomeric carbon configuration was encoded as “B” for β. This nomenclature is shown in Figure 10D.

The analysis of the available crystal structures showed that the inter-glycosidic torsions of GlcA*p* (1→3) GalNAc*p* (φ_H_ (O5–C1–O3′–C3′) and ψ_H_ (C1– O3′–C3′–C2′)) and GalNAc*p* (1→4) GlcA*p* (φ_H_ (O5–C1–O4′–C4′) and ψ_H_ (C1– O4′–C4′–C3′)) fell within a relatively narrow range and were essentially invariant, regardless of the substitution pattern [32]. Thus, average bond torsions were used (Table 4). Each disaccharide sequence was built and energy-minimized at the average Φ_H_ and Ψ_H_ values subject to a restraining force constant of 0.01 kcal^−1^mol^−1^·deg^−2^. The Sybyl atom types for sulfur and oxygen atoms in –SO_3_– groups were modified to S.o2 and O.co2, respectively, and the bond type between these atoms was set to ‘aromatic.’

The disaccharide building blocks were then used to build the desired library using SPL scripts, and then each sequence was minimized in an automated manner. Overall, the virtual library of tetrasaccharide sequences was combinatorially built from the 24 disaccharide building blocks and included a total of 192 sequences. Subsequent energy minimization was performed using the Tripos force field with Gasteiger–Hückel charges, a fixed dielectric constant of 80, and a non-bonded cutoff radius of 8 Å. Minimization was carried out for a maximum of 5000 iterations subject to a termination gradient of 0.05 kcal/(mol-Å). Likewise, a hexasaccharide library was generated and consisted of 1536 unique sequences.

### 3.3. Preparation of TGF-β2 Structure for Docking

The coordinates of the structure of human TGF-β2 (PDB ID: 1 TFG, 2.2 Å resolution) were taken from the Protein Data Bank (www.rcsb.org/pdb). The preparation of TGF-β2 structure was carried out using the “biopolymer protein preparation” module in SybylX, version 2.0 (Certara, St. Louis, MO, USA). First, the water molecules were stripped from the structure, ε nitrogens of His34 and His58 were set to their protonated forms, hydrogen atoms were added, and then the structure was minimized for 5000 iterations with a gradient of 0.05 kcal/(mol Å) using the Powell method. The potential sites of CS binding were identified by searching for the presence of the Cardin and Weintraub sequence motif (Appendix A) [34] and identifying electropositive domains through surface electrostatic potential calculations using the APBS tool in PyMol (https://www.pymol.org/). Two potential binding regions (Figure 2) were identified: BS1, with Lys25, Arg26, Lys31, His34, Lys37, Lys94, and Lys97, and BS2, with His58, Arg60, and Lys110.

### 3.4. Docking of Library of CS Sequences

The molecular docking of the library of CS sequences onto the structure of TGF-β2 was performed using GOLD v.5.6 [45], as described earlier in our works to understand protein–GAG interactions [23,24,25,26,27,28]. GOLD provides a range of scoring functions and customizable docking protocols that work for different GAG types and chain lengths. In a manner similar to our studies on the screening of Hp/HS libraries [23,26], the parameters for screening the CS sequences were optimized. The grid center was defined as the center of the residues in BS1 and BS2. The inter-glycosidic bonds were constrained within the normal range observed in nature. As the initial conformer population for each GAG sequence was selected at random, several genetic algorithm (GA) runs were required to more reliably predict bound conformations. The optimized parameters included 100 GA runs for each sequence docked onto a site of 12–16 Å radius (depending on the chain length and number of rotatable bonds) and 100,000 iterations. The algorithm evaluated GOLDScore and/or RMSD between top-ranked solutions on a continuous basis to identify the most optimal poses. GOLDScore is defined as the sum of HB_EXT_ and 1.375×VDW_EXT_, where the former corresponds to non-bonded inter-molecular H-bond forces and the latter corresponds to van der Waals forces, as reported earlier [23,26]. Collectively, the 100 GA runs formed one docking experiment from which the top two solutions were considered for further analysis. Experiments were minimally performed in triplicate, which yielded at least six solutions. To enhance efficiency, the GA was set to pre-terminate if the top two ranked solutions were within 2.5 Å RMSD. A one or two-step docking protocol was utilized depending on the library size, as shown in Figure 1.

### 3.5. Initial Preparations for Molecular Dynamics (MD)

The CVLS approach identified that TGF-β2 has two potential binding sites, namely BS1 and BS2. To understand the nature of interactions in solution, we carried out the MD simulations on CS chains binding independently at either BS1 or BS2. The initial structures for MD runs were taken from the CVLS output, which provided the docked complexes for the best sequence(s). The residue and atom labeling of the bound CS hexasaccharide (CS06) were altered to match the GLYCAM library as required (see http://glycam.org/docs/forcefield/glycam-naming-2/. Both the protein and the ligand were loaded into XLEAP of the AMBER14 suite. The glycosidic linkages and formal charge of CS06 were re-checked to ensure their appropriateness. Similarly, the protein structure loaded from the Protein Data Bank (PDB, www.rcsb.org was also checked for completeness, as expected. To bring the total charge of the complex to zero, the system was neutralized with the addition of appropriate number of counter ions (either Na^+^ or Cl^−^). Amber-ff14SB force field and GLYCAM_06j-1 force field parameters were used for protein and ligand preparation, respectively. This charge-neutralized complex was enveloped in a three-point water (TIP3P) molecule box with a minimum distance of 12 Å between the walls and any atom of the complex. The initial coordinates and parameters of the solvated protein–GAG complexes were saved before initializing MD runs. Each solvated protein–GAG complex was minimized in two steps with a 10 Å non-bonded cutoff. In the first step, the solute atoms were restrained with a force constant of 100 kcal/(mol. Å^2^), while the water molecules were relaxed using 500 cycles of the steepest descent method and 2000 cycles of the conjugate gradient method. In the second step, the whole system was relaxed using a conjugate gradient minimization of 2500 cycles without any restraints.

### 3.6. MD Simulations

Each solvated protein–GAG complex was equilibrated in three phases to achieve desired temperature and pressure with the integration step of 2 fs. In the first phase, the temperature was brought to 300 K using the temperature coupling with a time constant of 2 ps. In the second phase, the system was brought to a constant pressure using isotropic position scaling. Equilibration was carried out for 1 ns with initial strong restraints on the solute, which were systematically reduced. The production run was performed in an NPT ensemble with an integration time step of 1 fs. Bonds involving hydrogen atoms were constrained using the SHAKE algorithm. Maxwell distribution was used to assign the initial velocities. Each MD trajectory was computed for either 25 or 50 ns. Equilibration and simulation processes were validated by monitoring the physical observables of the system, including the energy (total, potential, and kinetic), temperature, and pressure as the function of the simulation time, which confirmed NPT ensemble settings (not shown).

### 3.7. Analysis of MD Simulations

Free energy calculations on TGF-β2–CS06 complexes were computed using the post-processing MM-PB(GB)SA method [50] from the MD trajectories. MM-GBSA employed SRED to estimate the energy contributions of each receptor residue in the bound state. Energy calculations were performed using the default parameter settings by employing the Python version of MM-PB(GB)SA module from Amber Tools13 (Case 2012) (refer to http://ambermd.org/tutorials/advanced/tutorial3/. Typically, these calculations were performed using the last 20 ns trajectory, which was represented by a total of 2000 structures.

## 4. Significance

This work presents an exhaustive computational simulation of the TGF-β2–CS interaction for the first time. In the process, we generated the library of all possible CS oligosaccharides from di- to hexa-saccharide chain lengths for the first time and demonstrated that the CVLS algorithm can be efficiently implemented for CS sequences. It is important to note that binding sites on proteins typically recognize GAG chains no longer than hexasaccharides [51]. This implies that the CVLS algorithm could be utilized for a broader application to understand CS recognition by multiple other proteins [23,24,25,26,27,28].

It is interesting that our computational results rigorously support the conclusion that TGF-β2 has two sites of binding for CS sequences. This in silico model is in agreement with prior biophysical studies that suggested a biphasic interaction model [21]. Though the biophysical studies were performed with heterogenous polymeric CS, it is likely that the polymer also engaged the sites identified for oligosaccharides in this study. However, whether the TGF-β2–CS complex (polymeric chain) has a stoichiometry of 1:1 or 1:2 remains unknown at this time. The current results showed that the two sites of binding, BS1 and BS2, are not linear (Figure 6, Figure 7 and Figure 8 and Appendix A). Though this favors the possibility of two different CS chains binding to TGF-β2, we cannot discount the possibility of one CS chain engaging both sites of binding.

Our CVLS study indicates that longer and more sulfated sequences are predicted to bind better than shorter and less sulfated CS sequences. This is in line with experiments reported in the literature [21]. However, a novel conclusion from our study is that the CS06 sequences identified as preferentially recognized by either BS1 or BS2 of TGF-β2 were generally higher sulfated oligosaccharides. For example, the majority of highly selective hexasaccharides contain six or seven sulfate groups (Table 3). Between the two sites, CS06 sequences with a preference for BS1 contained only four sulfate groups, while the majority of those with preference for BS2 contained six or more sulfates.

Highly sulfated CS sequences are not commonly found in nature. More importantly, the majority of highly selective sequences displayed either GlcA2S*p*–GalNAc4S6S*p* or GalNAc4S6Sp–GlcA2S*p* structures. Considering that GlcA2S*p* is a rare saccharide residue present in nature, these disaccharides represent highly rare structures preferred by TGF-β2. Additionally, both BS1- and BS2-specific sequences displayed preference for the rare GlcA2S*p* saccharide. This implies that common CS sequences would be expected to interact with TGF-β2 in a non-selective manner, i.e., the majority of sequences would not display a preference for either site of binding or differential interaction strengths.

Considering that common natural CS is expected to be non-selective, the CVLS results present an extremely interesting possibility of antagonizing this system. The hexasaccharide sequences identified to selectively bind in BS1 and/or BS2 would display higher binding affinities and thereby out-compete the common natural CS biopolymers. In fact, a dodecasaccharide that combines the BS1- and BS2-preferred sequences would exhibit a much higher antagonistic effect. Such a 12-mer may be very useful as a chemical probe to understand the role of TGF-β2 in various settings of cancer progression and metastasis. Unfortunately, synthesizing such a 12-mer is not an easy task.

Finally, we studied the TGF-β2 monomer in this work. Likewise, the dimer form should also be studied against the library of CS sequences. This work is currently in progress and will include exhaustive CVLS screening against the library of thousands of CS sequences, the assignment of selectivity, and the design of longer sequences that span the two sites of binding.

Overall, this work presents key principles of recognition regarding the TGF-β2–CS system. The in silico library of all possible CS oligosaccharides is now available for advanced studies on various protein–CS systems. Finally, the study led to the identification of unique CS sequences that are predicted to selectively recognize TGF-β2 and may out-compete common natural CS biopolymers.

## Figures and Tables

**Figure 1 ijms-22-07542-f001:**
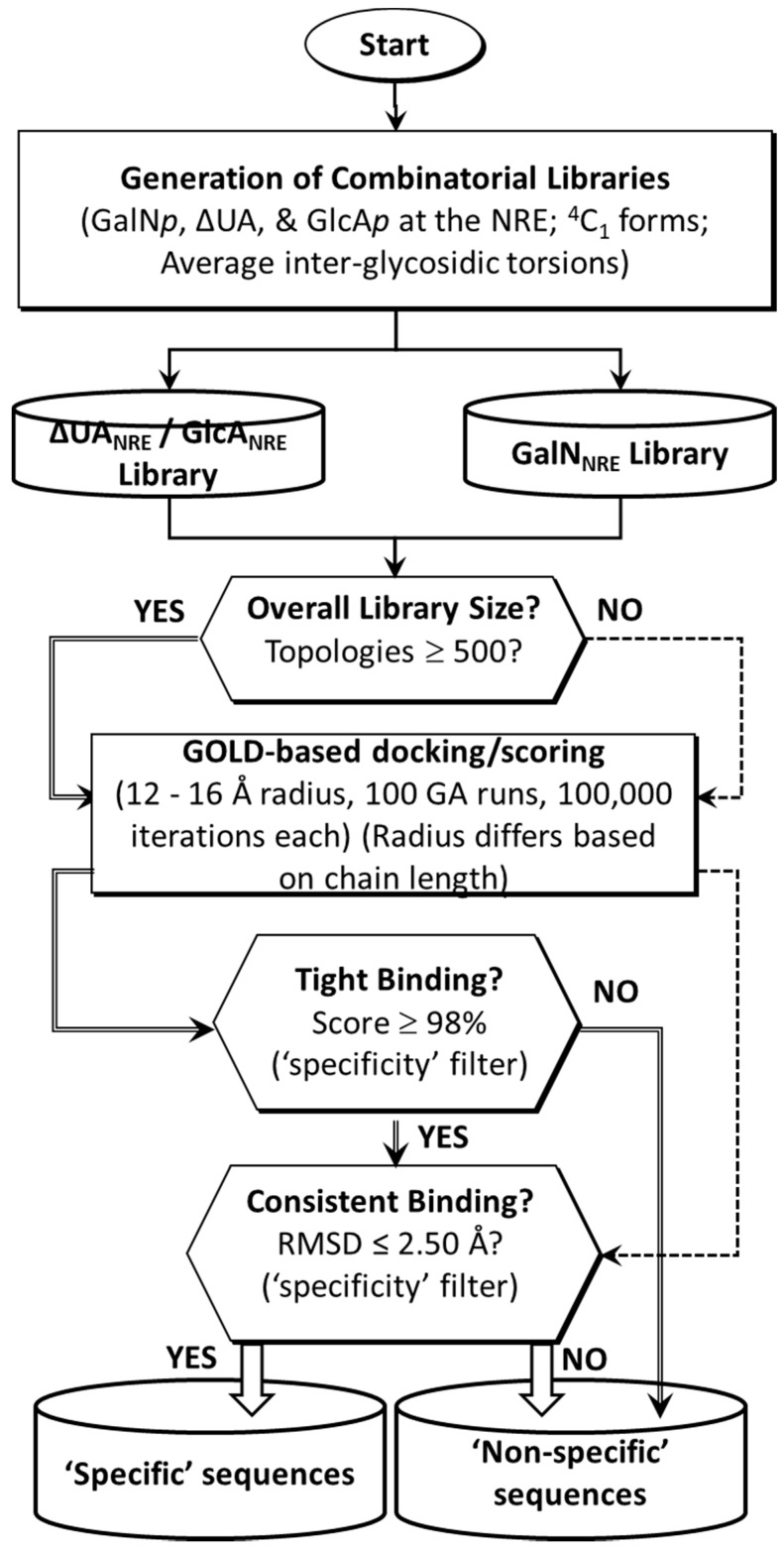
Combinatorial virtual library screening (CVLS) protocol used to study the TGF-β2–CS interaction. The CVLS protocol assessed the interaction of the CS02, CS04, and CS06 sequences using a dual-filter strategy that relied on GOLDScore and geometric convergence (RMSD) filters to assess the selectivity of binding.

**Figure 2 ijms-22-07542-f002:**
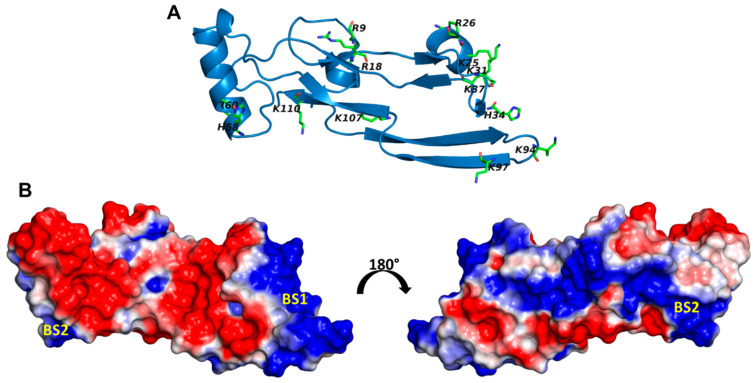
Electrostatic surface potential of TGF-β2: (**A**) the structure of transforming growth factor (TGF-β2) extracted from PDB (1TGF) showing the basic residues; (**B**) two potential GAG-binding sites (BS1 and BS2) identified based on electrostatic surface potential.

**Figure 3 ijms-22-07542-f003:**
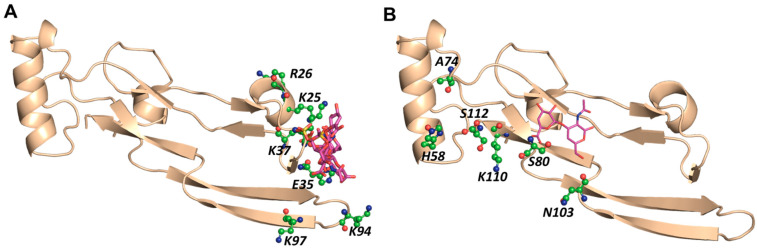
CVLS-predicted disaccharide sequences from the ΔUA_NRE_, GlcA_NRE_, and GlcN_NRE_ libraries of CS that prefer to bind to (**A**) BS1 and (**B**) BS2 of TGF-β2 (magenta color by atom sticks). The disaccharides do not display favorable high interaction scores.

**Figure 4 ijms-22-07542-f004:**
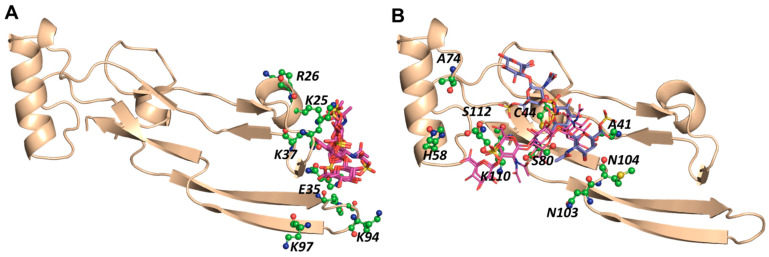
CVLS-predicted tetrasaccharide sequences from the ΔUA_NRE_, GlcA_NRE_, and GlcN_NRE_ libraries of CS that prefer to bind to (**A**) BS1 and (**B**) BS2 of TGF-β2 (magenta and blue colors by atom sticks, respectively). See text for details.

**Figure 5 ijms-22-07542-f005:**
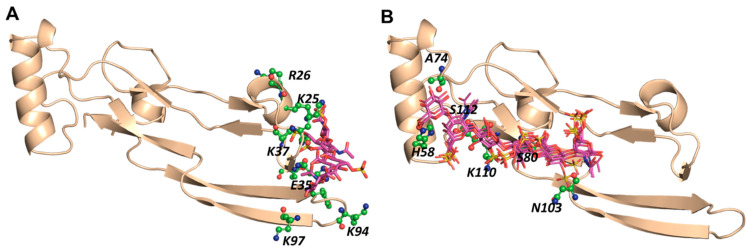
CVLS-predicted hexasaccharide sequences from the ΔUA_NRE_, GlcA_NRE_, and GlcN_NRE_ libraries of CS that prefer to bind to (**A**) BS1 and (**B**) BS2 of TGF-β2 (magenta and blue color by atom sticks, respectively). See text for details.

**Figure 6 ijms-22-07542-f006:**
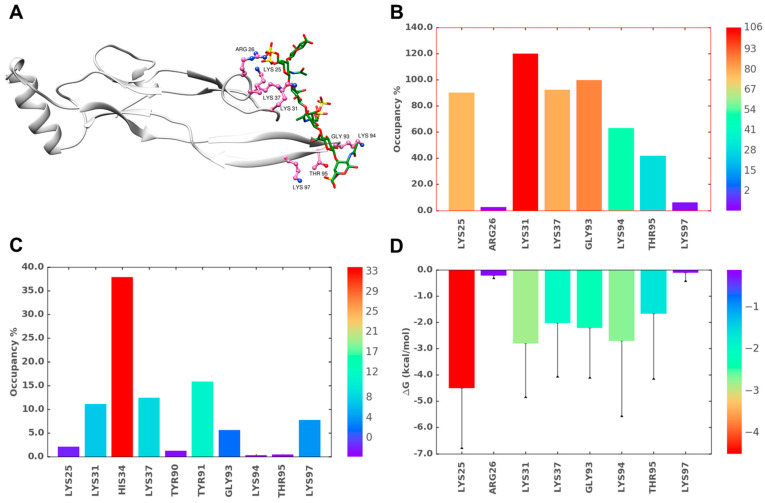
MD simulation of the TGF-β2–CS06 interaction at binding site I (BS1). (**A**) A representative structure from the MD trajectory showing the hydrogen bonds (H-bonds) between TGF-β2 residues (hot pink; ball and stick representation) and CS06 (green sticks). The protein ribbon is shown in light grey. (**B**) Inter-molecular H-bond formation in terms of % occupancy (proportion of MD frames showing direct H-bonds) for residues of BS1. (**C**) Proportion of water-mediated H-bonds (i.e., occupancy) between TGF-β2 for residues in BS1 and CS06. (**D**) Single-residue energy decomposition (SRED) values for the TGF-β2–CS06 complex for residues of BS1. Error bars show standard deviation. For ease of interpretation, values from lower to higher are represented by rainbow color (blue→red).

**Figure 7 ijms-22-07542-f007:**
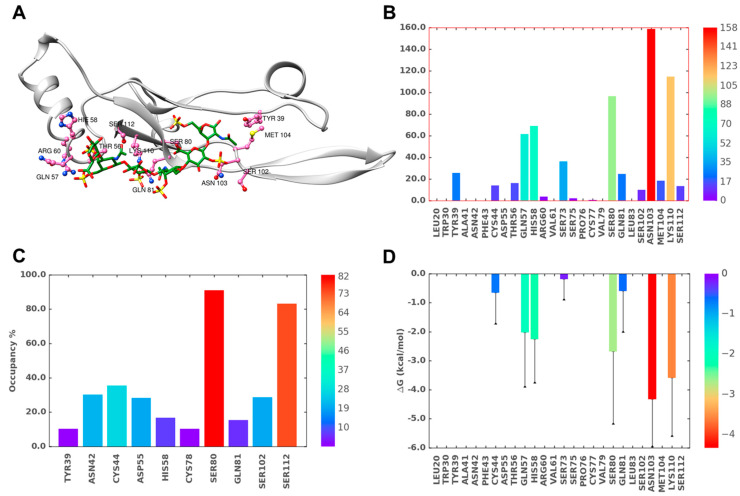
MD simulation of the TGF-β2–CS06 interaction at binding site 2 (BS2). (**A**) A representative structure from the MD trajectory showing the hydrogen bonds (H-bonds) between TGF-β2 residues (hot pink; ball and stick representation) and CS06 (green sticks). The protein ribbon is shown in light grey. (**B**) Inter-molecular H-bond formation in terms of % occupancy (proportion of MD frames showing direct H-bonds) for residues of BS2. (**C**) Proportion of water-mediated H-bonds (i.e., occupancy) between TGF-β2 for residues in BS2 and CS06. (**D**) Single-residue energy decomposition (SRED) values for the TGF-β2–CS06 complex for residues of BS2. Error bars show standard deviation. For ease of interpretation, values from lower to higher are represented by rainbow color (blue→red).

**Figure 8 ijms-22-07542-f008:**
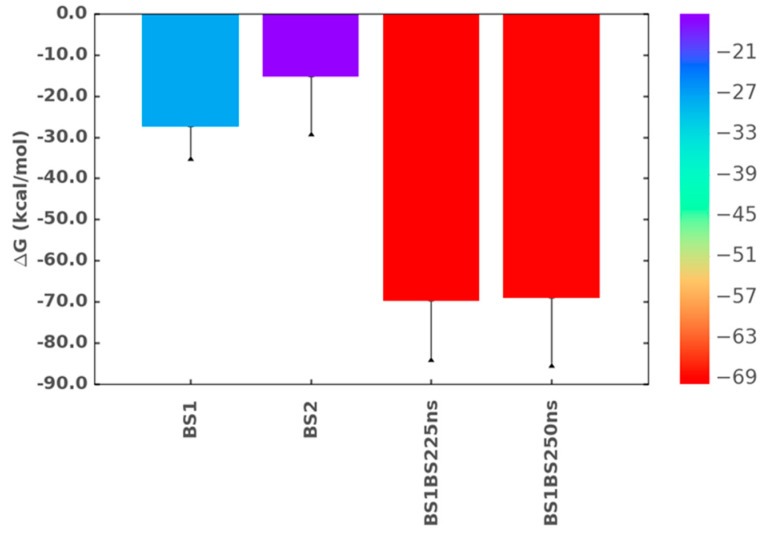
Free energy of binding for the TGF-β2–CS06 complex when ligand binding occurred only in BS1 or BS2 or simultaneously in BS1 and BS2 using MM-GBSA studies. The free energy of binding significantly increased when both sites of binding were occupied by two CS06 sequences at the same time (red color). The free energy value remained unchanged when the MD simulation time was increased from 25 to 50 ns. Error bars show standard deviation. For ease of interpretation, values from lower to higher are represented by rainbow color (blue→red).

**Figure 9 ijms-22-07542-f009:**
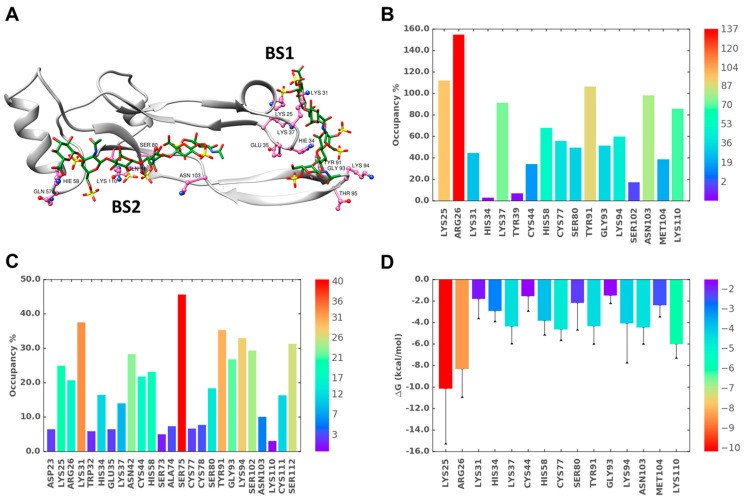
MD simulation of the TGF-β2–CS06 complex when engaged at both sites of binding, BS1 and BS2. (**A**) A representative structure from the MD trajectory showing the hydrogen bonds (H-bonds) between TGF-β2 residues (hot pink; ball and stick representation) and CS06 (green sticks). The protein ribbon is shown in light grey. (**B**) Inter-molecular H-bond formation in terms of % occupancy (proportion of MD frames showing direct H-bonds) for residues of BS1 and BS2. (**C**) Proportion of water-mediated H-bonds (i.e., occupancy) between TGF-β2 for residues in BS1 and BS2 and CS06. (**D**) Single-residue energy decomposition (SRED) values for the TGF-β2–CS06 complex for residues of BS1 and BS2. Error bars show standard deviation. For ease of interpretation, values from lower to higher are represented by rainbow color (blue→red).

**Figure 10 ijms-22-07542-f010:**
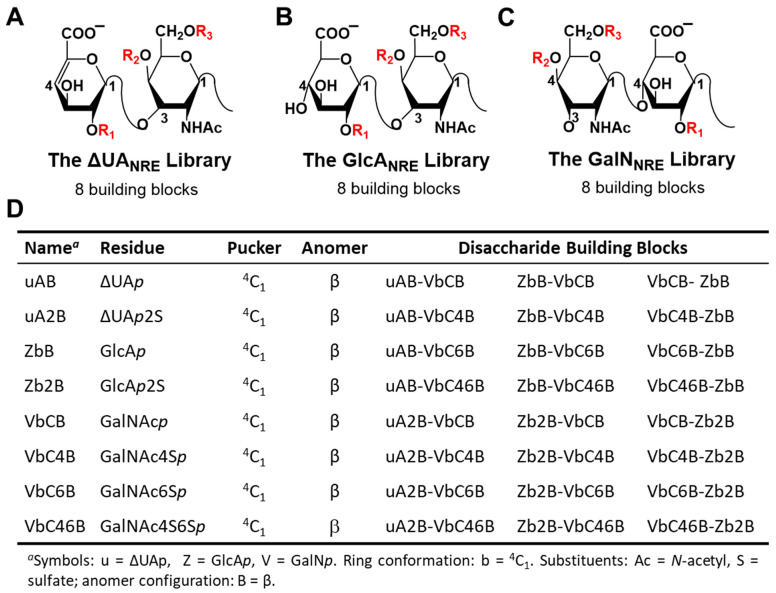
The chondroitin sulfate (CS) disaccharide building blocks and their naming conventions: (**A**) the ΔUA_NRE_ library of oligosaccharides (di- to hexasaccharide) has an unsaturated uronic acid residue at the non-reducing end (NRE) and a GalN*p* residue at the reducing end; (**B**) the GlcA_NRE_ library has a GlcA*p* residue at the NRE, and (**C**) the GalN_NRE_ library has a GalN*p* residue at the NRE; (**D**) the list of monomer residues and the R_1_, R_2_, and R_3_ variations for ΔUA*p*, GlcA*p,* and GalN*p* considered in this study. These variations lead to 24 disaccharide building blocks.

**Table 1 ijms-22-07542-t001:** CS disaccharide sequences from the library of the 24 sequences that satisfied the dual-filter CVLS strategy for TGF-β2.

Binding Site	Disaccharide Sequence	RMSD (Å)	No of Sulfates	GOLDScore
1	uAB–VbC4B	2.5	1	53.1
	VbC6B–Zb2B	2.5	2	56.3
	ZbB–VbCB	2.5	0	44.2
2	uAB–VbCB	2.3	1	39.2

**Table 2 ijms-22-07542-t002:** CS tetrasaccharide (CS04) sequences from the library of 192 sequences that satisfied the dual-filter CVLS strategy for TGF-β2.

Binding Site	Tetrasaccharide Sequence	RMSD (Å)	No of Sulfate	GOLDScore
1	Zb2B–VbC4B–Zb2B–VbC6B	1.5	4	61.7
	uAB–VbC4B–Zb2B–VbCB	1.9	2	50.3
	ZbB–VbC4B–ZbB–VbCB	2.3	1	45.1
2	uAB–VbC6B–Zb2B–VbC46B	2.5	4	50.6
	VbC6B–ZbB–VbC46B–ZbB	1.8	3	53.9
	VbC6B–ZbB–VbC4B–ZbB	1.3	2	52.5
	VbCB–Zb2B–VbC6B–Zb2B	2.0	3	53.8
	VbCB–ZbB–VbC4B–ZbB	1.0	1	48.1
	VbCB–ZbB–VbCB–ZbB	1.6	0	46.5
	ZbB–VbC4B–ZbB–VbC4B	1.7	2	57.8

**Table 3 ijms-22-07542-t003:** CS hexasaccharide (CS06) sequences from the library of 1536 sequences that satisfied the dual-filter CVLS strategy for TGF-β2.

Binding Site	Hexasaccharide Sequence	RMSD (Å)	No of Sulfates	GOLDScore
1	ZbB–VbC4B–Zb2B–VbC4B–ZbB–VbC6B	2.5	4	87.6
2	Zb2B–VbC6B–Zb2B–VbC4B–ZbB–VbC46B	1.8	6	85.8
	Zb2B–VbC6B–ZbB–VbC46B–Zb2B–VbC6B	2.0	6	79.0
	ZbB–VbC46B–Zb2B–VbC46B–Zb2B–VbC4B	2.2	7	71.8
	ZbB–VbC46B–Zb2B–VbC4B–Zb2B–VbC46B	1.6	7	82.7
	ZbB–VbC46B–Zb2B–VbC4B–Zb2B–VbC6B	1.6	6	88.4
	ZbB–VbC46B–Zb2B–VbCB–Zb2B–VbC46B	2.2	6	86.7
	ZbB–VbC46B–Zb2B–VbCB–ZbB–VbC6B	2.1	4	71.7
	ZbB–VbC46B–ZbB–VbC46B–ZbB–VbC4B	1.8	5	70.3

**Table 4 ijms-22-07542-t004:** Average torsion across the 1→3 and 1→4 inter-glycosidic bonds used in CVLS.

Disaccharide Building Blocks	φ	ψ	Atoms
GlcA*p*(1→3)GalNAc*p*	−81	−129	O5–C1–O3′–C3′ (φ) C1–O3′–C3′–C2′ (ψ)
GalNAc*p*(1→4)GlcA*p*	−69	131	O5–C1–O4′–C4′ (φ) C1–O4′–C4′–C3′ (ψ)

## Data Availability

The libraries of CS sequences were generated using licensed software SYBLYLX. These are available to any user upon request.

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
