# Peer review of "Combinatorial Virtual Library Screening Study of Transforming Growth Factor-β2–Chondroitin Sulfate System"

_ijms, 2021, doi:10.3390/ijms22147542_

Round 1

Reviewer 1 Report

This article is a report of computational studies of the interaction between TGFbeta-2 and chondroitin sulfate oligosaccharides, with the aim of identifying specific monosaccharide sequences with high affinity and specificity of binding. It is also the first presentation of the virtual library of small CS oligosaccharides developed by this group following on from their successful applications of a similar library of heparin/HS structures.

The manuscript is well written, describing the methods used clearly and presenting results that indicate the presence of two distinct binding sites on the TGFbeta-2 monomer surface. However, it is not clear why the authors concentrate on the monomeric form of TFGbeta-2 as this protein when mature is said to exist as a covalently linked dimer, as shown in crystal structures such as 1TFG (SS bond at Cys77; see Fig 1 of reference 10). The study must be extended to include docking to the dimer, with MD of the best results. The significance of the results for biology depends on this. A quick look at 1TFG and 2TGI indicates that BS1 is well exposed in the dimer, but BS2 might overlap with the dimerization face.

This is extra work but it needs to be done.

Author Response

We appreciate the reviewer's comment that the dimer should also be studied against the library of CS sequences. Unfortunately, this implies that we will have to combine our second full manuscript into one massive publications. The principles enunciated in the work on dimer are also highly interesting and valuable just as the principles enunciated in the work on the monomer. This is the reason our work will have to be described in two papers. The paper on the dimer will include screening against the library of thousands of CS sequences, the assignment of selectivity, the design of longer sequences that span two sites of binding and much more. This will be coming soon. To partially address the reviewer's point, we have included few statements in the conclusion section to alert an interested reader to this soon-to-be published work.  

Reviewer 2 Report

The authors use computational virtual library screening combined with Molecular dynamics for the study of the interaction between TGF beta2 and Chondroitin Sulfate, using virtual libraries of di, tetra and hexasaccharides of CS. The hexasaccharides are more specific and the authors propose virtual models for the active complex. They have found two binding sites in TGF for sulfates CS hexasaccharides.

It is complicated to connect the number of sulfates in the hexasaccharide with the active sequences. Could you find a better way?

Author Response

It is not clear what the reviewer is intending to ask; however, we interpreted the question to imply whether higher number of sulfates correspond to higher selectivity. This is not to be found because the interaction appears to be fairly selective. We have added a few lines to add this point to the Results section. 

Round 2

Reviewer 1 Report

The authors have explained that a second manuscript will be published concerning CS binding to the TGF-beta2 dimer. This is now made clear in the Discussion. The authors should still make it clear from the abstract onwards in this paper that they are screening the monomer, not the dimer, for CS binding. Two simple edits should be added:

  1. In the Abstract line 24: add the word 'monomer' to read 'TGFbeta2 monomer'
  2. Similarly in the Introduction line 74: add the word 'monomer' to read 'TGFbeta2 monomer'

Author Response

The two changes have been made.